# Unboxing the Black Box of Attention Mechanisms in Remote Sensing Big Data Using XAI

**Erfan Hasanpour Zaryabi [1], Loghman Moradi [1], Bahareh Kalantar [2,*], Naonori Ueda [2] and Alfian Abdul Halin [3]**

[1] School of Surveying and Geospatial Engineering, College of Engineering, University of Tehran, Tehran 14648-54763, Iran

[2] RIKEN Center for Advanced Intelligence Project, Goal-Oriented Technology Research Group, Disaster Resilience Science Team, Tokyo 103-0027, Japan

[3] Department of Multimedia, Faculty of Computer Science & Information Technology, Universiti Putra Malaysia, Serdang 43400, Malaysia

* Correspondence: bahareh.kalantar@riken.jp; Tel.: +81-362252482

**Abstract:** This paper presents exploratory work looking into the effectiveness of attention mechanisms (AMs) in improving the task of building segmentation based on convolutional neural network (CNN) backbones. Firstly, we evaluate the effectiveness of CNN-based architectures with and without AMs. Secondly, we attempt to interpret the results produced by the CNNs using explainable artificial intelligence (XAI) methods. We compare CNNs with and without (vanilla) AMs for buildings detection. Five metrics are calculated, namely F1-score, precision, recall, intersection over union (IoU) and overall accuracy (OA). For the XAI portion of this work, the methods of Layer Gradient X activation and Layer DeepLIFT are used to explore the internal AMs and their overall effects on the network. Qualitative evaluation is based on color-coded value attribution to assess how the AMs facilitate the CNNs in performing buildings classification. We look at the effects of employing five AM algorithms, namely (i) squeeze and excitation (SE), (ii) convolution attention block module (CBAM), (iii) triplet attention, (iv) shuffle attention (SA), and (v) efficient channel attention (ECA). Experimental results indicate that AMs generally and markedly improve the quantitative metrics, with the attribution visualization results of XAI methods agreeing with the quantitative metrics.

**Keywords:** XAI; explainability; interpretability; attention mechanisms; building segmentation; CNN

## 1. Introduction

The field of aerospace remote sensing (RS) heavily relies on high-quality imagery data. For the most part, the data used are high-resolution remotely sensed images that are commonly collected and processed using advanced sensor and drone technology, coupled with digital image processing. It is then customary to feed these data to Artificial Intelligence (AI) algorithms for higher-level RS analysis [1,2]. Two AI subsets commonly come to mind, namely supervised Machine Learning (ML) and Deep Learning (DL). Utilizing both or either one of these technologies affords RS researchers semi or full automation in performing analysis and/or decision making for tasks such as land cover monitoring and categorization [3–5], damage detection/assessment [6–8], disaster planning and prediction [9–11], and river meandering analysis [12]. The AI methods, however, are only as good as the input provided. In RS research, as every pixel detail is crucial, the technologies collecting imagery data must be robust and reliable to feed the ML/DL engines.

Assuming quality data, RS researchers have used 'traditional' ML methods for RS tasks. For example, the support vector machine (SVM) and ensemble classifiers (e.g., Random Forest (RF) and gradient boosting) have been popular for RS classification, categorization, and detection [13,14]. In recent years, attention has shifted to DL algorithms such as convolutional neural networks (CNN) [15], graph neural networks (GNN) [16], and

generative adversarial networks (GAN) [17–19]. DL became popular due to the potential (and achievable) high accuracy of the algorithms, providing there are enough data and algorithms for training/testing.

One lingering issue in RS (or other image-related) research is dealing with problematic objects of interest, such as buildings and vehicles. Often times, these objects are too small or they are blocked/occluded by other objects, and the challenge is compounded with moving objects in video [20,21]. ML and DL algorithms can, to a certain extent, detect some of the objects. However, consistent detection is challenging when using, for example, a vanilla CNN, or even when using established object detectors such as YOLO [22] or region-based CNNs (RCNN).

Due to the challenge in constant detection, tracking, and extraction of such objects, recent algorithms have focused on attention mechanisms (AMs). In computer vision, AM is when the algorithm focuses only on specific or more important parts of an image, while ignoring irrelevant parts. For example, when 'looking' at an image of a dog, the undivided focus should be on the dog itself instead of focusing on the background. AM basically attempts to imitate the human brain-eye vision system, by highlighting the most important information instead of paying equal attention to other less important information. The earliest use of AM was in natural language processing (NLP), in which AMs guided deep neural networks (DNNs) to more accurately highlight prominent details of important speech features while suppressing irrelevant features [23]. This opened the door for AM applications in other fields, in which it is now commonly used in DL-based methods to increase accuracy.

Additionally, the large numbers of parameters in some DL models make them complex and harder to interpret. In such cases, even a slight dysfunction of a DL algorithm can have catastrophic consequences. For instance, an autonomous car with a minor computer vision component dysfunction may lead to a fatal accident. Regardless of excellent validation and even unseen data metrics, DL models could inherently fail to learn representations from the data that a human might consider important. Explaining the decisions made by DNNs require knowledge of the internal operations of DNNs. To do this, explainable artificial intelligence (XAI) methods could provide human-interpretable explanations for better understanding ML/DL "*black-box*" decisions.

To the best of our knowledge, the effectiveness of the *black box* of AMs has not yet to be explored. We intend to fill this gap by investigating the effectiveness of AMs for RS-related tasks, based on XAI. To that end, we look at two subsets of evaluation metrics. Firstly, we examine the results of CNN methods, with and without AMs, from the quantitative metrics of accuracy, F1-score, precision, etc. Secondly, we look at the AMs' effectiveness using two qualitative XAI methods: (i) Layer Gradient X activation, and (ii) Layer DeepLIFT [24]. These are used to explore internal AMs and their overall effects on the respective network. The domains of interest focused on are building information extraction and segmentation, as they are not only most applicable, but are also crucial RS task for urban area analysis and development. Furthermore, other RS tasks rely on these two domains, namely 3D modeling, smart city planning, population estimation, and disaster management.

Due to this, we develop and evaluate our building segmentation framework with AMs and XAI. Two different CNN backbones are utilized, i.e., SegNet [25] and UNet [26]. Five AM algorithms are considered as attention blocks, namely (i) squeeze and excitation (SE) [27], (ii) convolution attention block module (CBAM) [28], (iii) triplet attention [29], (iv) shuffle attention (SA) [30] and (iv) efficient channel attention (ECA) [31]. Evaluation is based on the xBD dataset [32], which contains a large number of high-quality satellite RS building imagery, along with building information.

In summary, our work offers the following contributions:

(1) We propose an encoder-decoder CNN framework for building segmentation;
(2) We evaluate the effectiveness of the AMs in each RS related task (based on the aforementioned metrics and XAI layer contribution methods);

(3) We provide interpretations of attention blocks in different layers of the framework;
(4) We attempt to unbox the *black box* of the AMs on the model decision by using XAI layer contribution methods.

Going forward, Section 2 reviews DL attention-based methods, and XAI in different RS scenarios. A comprehensive explanation of the developed framework, AMs and XAI methods is then given in Section 3. Dataset information, experimental results and discussions are presented in Section 4. In Sections 5, we provide concluding remarks with regards to our findings.

## 2. Related Studies

### 2.1. Attention Mechanisms in RS Domains

In this section, we cover recently published articles in the field of RS applications employing AMs. Specifically, consistent with Ghaffarian et al. [33] assertion of their importance in RS research, we review image classification, image segmentation, object detection, change detection, and building extraction.

### 2.1.1. Image Classification

In recent years, several AMs have been included in several CNN architectures. Two notable works were on scene classification, by Alhichri et al. [34] and Tong et al. [35]. The former proposed the EfficientNet-B3-Attn-2 CNN scene classifier, with a custom AM. The authors concluded that the AM effectiveness depended on where it was used in the model. The latter proposed a channel attention-based CNN with DenseNet121 as the backbone, in which scene classification accuracy with channel attention showed a general increase by ~1%. Ma et al. [36] proposed a spatial spectrum collaborative network (SCCA-Net), utilizing spatial and channel attention on panchromatic and multi-spectral images. Their method managed to increase the accuracy for multi-resolution image classification. In hyperspectral image classification, Li et al. [37] proposed a mechanism based on dual-attention, which reduced the number of training samples during training. Specifically, double branch dual-attention is used for highlighting spectral and spatial features. In addition, a channel and spatial attention blocks are applied to the branches to optimize feature extraction. Zhu et al. [38] performed hyperspectral image classification using a channel-spatial attention module by utilizing raw 3D data as an input. Channel attention is performed to select useful spectral bands and spatial attention for selecting useful spatial information. They also included a spectral-spatial attention module to refine the learned features.

### 2.1.2. Image Segmentation

For area segmentation, Zhao et al. [39] proposed an encoder-decoder attention-based semantic segmentation network. ResNet101 is the backbone for the encoder part, which extracts detailed features, aided by the pyramid attention pooling (PAP) module to capture multi-scale context. For the decoder, they implemented an attention up-sampling module to recover additional information from the shallow layer. Nie et al. [40] proposed a segmentation-based ship detection framework using an improved Mask R-CNN. A feature pyramid network (FPN) is used to obtain the feature pyramid, and a bottom-up path was added to shorten the information path between the lower and top layers. Channel and spatial AM were used in the bottom-up path. In this work, the spatial attention increased ship detection and segmentation accuracies by 4.9% and 3.2%, respectively, and the channel-wise attention ship detection and segmentation accuracies improved by 4.3% and 2.9%, respectively. Ma et al. [41] proposed a super-pixel segmentation in SAR (Synthetic Aperture Radar) data using an attention graph convolution network (AGCN) comprising an AM and graph convolution networks (GCN). Here, the AM is meant to assist the graph convolution layer to focus on only the important and relevant nodes. Li et al. [42] proposed a dual path attention network (DPA-Net) for image segmentation. Two at-

tention modules, focusing on spatial and channel information, were added to the segmentation model to increase the network's ability for feature extraction. Ding et al. [43] implemented a patch-wise local attention module to enhance semantic segmentation for RS images. Their model operates on feature maps and aggregating context information. An attention-embedding module is also implemented to enhance the semantic representation of low-level features.

### 2.1.3. Object Detection

DL-based object detection, aided by AM, can also be used for monitoring and discovery tasks. For example, Li et al. [44] proposed an object detection network consisting of (i) a refine feature pyramid network (RFPN) and (ii) a multi-layer attention network (MANet). The RFPN is responsible for generating a set of multi-scale feature maps whereas the MANet magnifies (highlights) target features while suppressing noise. It is worth noting that in the MANet component, each attention layer consists of position and channel attention blocks. Zhao et al. [45] and Zhou et al. [46] performed the task of transport detection. The former made use of CBAM [28] for ship detection in SAR images, while the latter proposed a local attention network to improve the detection of occluded airplanes with a standard feature pyramid network and a local attention block.

### 2.1.4. Change Detection

AM can also be applied for change detection, where the focus is recognizing any changes to an object or area of interest. This is often carried out using multi-temporal RS image sequences that may vary through time. Change detection commonly entails assigning a binary label to each pixel, based on multi-temporal co-registered data. Jiang et al. [47] proposed a building change detection Siamese network with an encoder-decoder structure. They utilized the VGG16 [48] network as the backbone to extract features from image pairs. A co-attention module was used at the end of the encoder to learn the correlations between the features of image pairs. Jie et al. [49] proposed a change detection method based on a convolutional Siamese network with a dual attention module to improve feature discrimination by highlighting useful information. Song et al. [50] proposed an attention-guided network for building change detection in high-resolution RS images. Their spatial attention and dual channel-wise attention modules improved F1-score and IoU (intersection over union) by 1.48% and 2.33%, respectively. In [51], by combining CNN and bidirectional long short-term memory network, along with soft AM (which assigns corresponding attention weight to image features), the authors showed improvement in change information of the image. Chen et al. [52] proposed a Unet-based encoder-decoder network based on the residual attention module. A new attention mask structure was also implemented for feature extraction only from regions that exhibited obvious change.

### 2.1.5. Building Extraction

Building extraction from RS images provides crucial information for a wide range of applications pertaining to disaster management, infrastructure planning, demography information, and 3D modeling. Zhang et al. [53] proposed a hybrid attention-aware fusion network (HAFNet) for building extraction by combining high-resolution images and Li-DAR (Light Detection And Ranging) data. In this work, two networks extract and separately learn features from both types of data. Another network is designed to investigate cross-model complements by fusing the other two network features. Ultimately, the predictions of all three networks are fused to produce building extraction results. In addition, an attention block was used to highlight useful features and suppress irrelevant features. Lo et al. [54] proposed a two-stage Unet-based framework for farmland building extraction. The first stage performed building extraction using a ResNet as the basic structure, a spatial-channel AM, and a multi-scale fusion module where results are predicted from fused data. The second stage is boundary optimization and fusion processing, wherein

the morphological image processing operations of opening and closing are used for image optimization and merging. Guo et al. [55] also used Unet where they implemented a multi-loss network for building extraction by using an attention block. The attention block enhances the sensitivity of feature information extraction and suppressing the effect of background information. Zhou et al. [56] proposed a two-part building extraction network. The first part is the backbone of the network, which performs building feature extraction. The second part is the pyramid self-attention (PISA) module, which obtains global and comprehensive features for the buildings. Pan et al. [57] used an unsupervised approach, wherein a GAN (generative adversarial network) was implemented for building extraction. Spatial and channel AMs are introduced to the network to highlight the useful information. Notably, the channel attention SE block is adopted as channel attention. Chen et al. [58] used a self-attention module for building extracting, combined with a reconstruction-bias module in RS images. Tian et al. [59] proposed an encoder-decoder architecture for building extraction. In their work, atrous convolution, deformable convolution, and AM are combined for extracting features in the encoding path. They used CBAM [28] as the attention module. Das et al. [60] created a network termed as refined cross AttentionNet (RCA-Net) for extracting building footprints using pre-trained layers of the ECA-Net [31] and ResNet-50 [61]. They used global attention fuse (GAF) module to capture local and global cross-channel relationships.

### 2.2. XAI Applications in RS Domains

XAI involves processes/methods that aid humans in understanding the outputs of AI models. XAI is meant to facilitate the characterization of AI-enabled decision-making systems in terms of model metrics, fairness, and results/outcomes. In RS, XAI method is not yet widespread. We, however, believe models, especially those adopting ML and DL, need to be scrutinized so that more light can be shed on how decisions are made by the system/model. This will aid in avoiding the black box problem and also in future model development for various RS-related tasks.

Among the earlier works that attempt to scrutinize model results was that of Mandeep et al. [62]. They proposed a SAR target recognition framework based on CNN for image classification. The authors attempted to interpret the model using the local interpretable model agnostic explanation (LIME). Su et al. [63] investigated several XAI methods for the predictions of the CNN SAR images classifier network. They adopted two quantity evaluation methods for overall performance evaluation. Abdollahi et al. [64] used SHAP (SHapley Additive exPlanations) to rank input parameters and select appropriate features for their vegetation mapping network. In another study, a DNN was used to classify MODIS imagery of vegetation and metrological data to forecast wheat yield in [65]. The DNN was integrated with the regression activation maps (RAM) for interpreting the output of the model. Al-Najjar et al. [66] used SHAP to explore the impact of SAR and normalized difference vegetation index (NDVI) time-series features and ranked appropriate features for their landslide susceptibility mapping. Kakogeorgiou et al. [67] compared different XAI methods for satellite multispectral imagery classification.

From all these previous works, we observe that none has focused on AM in interpreting model results. Since AMs have been a crucial part of artificial intelligence (AI) development since 2017, we expect its adoption to increase in the foreseeable future. Therefore, model interpretation using XAI will also be necessary. Our work there intends to fill this gap, where we posit there is still lack of comprehensive work in evaluating the performance and effect of AMs and explaining their black box in RS-related ML/DL tasks. In this regard, we select the most well-known and cited AMs (SE, CBAM, ECA, SA, Triplet) in RS fields to explore the effectiveness of AMs, as well as to interpret the models in which they are implemented, by using XAI. In this article, we focus the RS domain specifically to the task of building segmentation. This task is selected since it is a crucial step in most RS image applications, such as 3D modeling, urban management, and especially damage assessment.

### 3. Methodology

Figure 1 illustrates the bespoke framework used to evaluate the effects of AMs (on the RS images) for building segmentation. The framework's architecture is based on the symmetric CNN model structure with two backbones, five AMs (SE, CBAM, ECA, SA, Triplet), and two XAI methods (Layer Gradient X Activation, Layer DeepLIFT). The central core of this framework is the decoder, which uses attention blocks and explainable gates to qualitatively and quantitatively interpret the impact of AMs on different backbones.

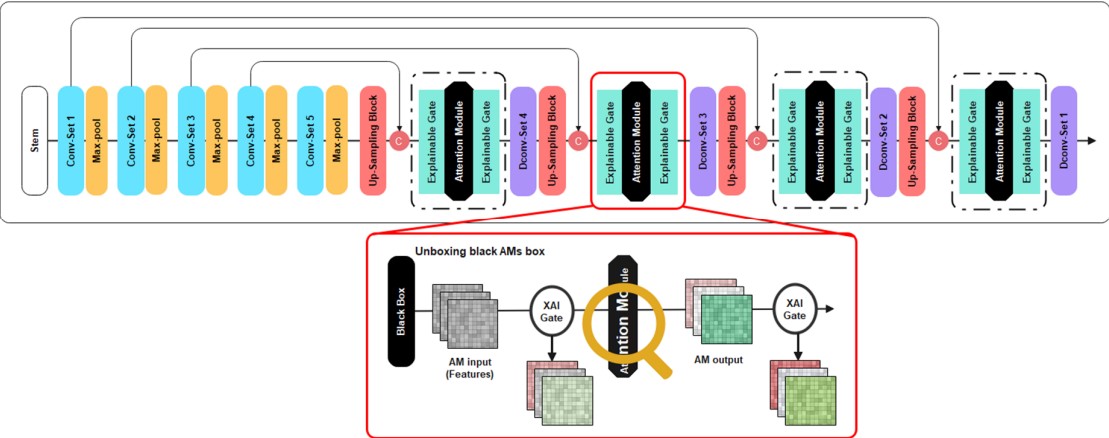

**Figure 1.** Overall framework of the proposed method. In the decoder part, four attention blocks and eight explainable gates are designed to evaluate the effect of AMs on different backbones and explain their black box.

The proposed framework is designed to learn the extraction of the building objects, while ignoring other/unimportant objects. The framework's symmetrical architecture includes the encoder and decoder parts; the encoder will extract multi-level features and condense the spatial data representation into a lower dimension, while the decoder converts this lower-dimensional representation back to the original dimension to reconstruct the input data. The encoder consists of five layers, and each layer includes a Conv-set block to extract deep-features and a max pooling to reduce the input size. The Conv-set blocks are designed according to the backbones, including SegNet and Unet encoders. Both SegNet and Unet have five layers in their encoders and each of these Conv-Set blocks is equal to its corresponding layer in the SegNet or Unet encoders. On the other hand, the decoder comprises four Deconv-set and up-sampling blocks, which is meant to recover the original input size. Also included are four attention blocks, so that focus is only on placed on significant parts of the input features. Since the framework uses two different backbones with the same decoder, a convolution filter is used in the up-sampling block to arrange the number of channels with respect to the backbone after input feature up-sampling. In general, after each up-scaling layer, several successive convolution layers are used to assemble a more precise output, whereby the Deconv-Set blocks, which consist of two convolution filters, are responsible for this role. Immediately after each up-sampling block, the features are passed from the encoder (with skip connections) and are concatenated with the up-scaled features. Since the up-sampling block and concatenation process increase input-feature vector dimension, using an attention block before each Deconv-set helps the Deconv-set block to reconstruct the output features (by highlighting significant parts of the features,). Therefore, attention blocks are placed before each Deconv-Set to make their effect more pronounced compared to a vanilla network (i.e., the framework without attention blocks). Details of the proposed framework are presented in Table 1.

**Table 1.** Parameters and feature size of the proposed framework. The feature size shows the framework has symmetrical architecture and attention blocks do not change feature size.

| Backbone | Encoder Layer | Feature Size | Kernel Size | Decoder Layer | Feature Size | Kernel Size |
|---|---|---|---|---|---|---|
| - | Input Image | $512 \times 512 \times 3$ | - | Up-Sampling Block Deconv-Set 4 | - $64 \times 64 \times 512$ | $3 \times 3$ |
| | Conv-Set 1 | $512 \times 512 \times 64$ | $3 \times 3$ | | | |
| | Conv-Set 2 | $256 \times 256 \times 128$ | $3 \times 3$ | Attention Block 4 | $64 \times 64 \times 512$ | $3 \times 3$ |
| SegNet | Conv-Set 3 | $256 \times 256 \times 128$ | $3 \times 3$ | Up-Sampling Block Deconv-Set 3 | - $128 \times 128 \times 256$ | $3 \times 3$ |
| | Conv-Set 4 | $64 \times 64 \times 512$ | $3 \times 3$ | Attention Block 3 | $128 \times 128 \times 256$ | $3 \times 3$ |
| | Conv-Set 5 | $32 \times 32 \times 1024$ | $3 \times 3$ | Up-Sampling Block Deconv-Set 2 | - $256 \times 256 \times 128$ | $3 \times 3$ |
| | Conv-Set 1 | $512 \times 512 \times 64$ | $3 \times 3$ | Attention Block 2 | $256 \times 256 \times 128$ | $3 \times 3$ |
| | Conv-Set 2 | $256 \times 256 \times 128$ | $3 \times 3$ | Up-Sampling Block Deconv-Set 1 | - $512 \times 512 \times 64$ | $3 \times 3$ |
| Unet | Conv-Set 3 | $256 \times 256 \times 128$ | $3 \times 3$ | Attention Block 1 | $512 \times 512 \times 64$ | $3 \times 3$ |
| | Conv-Set 4 | $64 \times 64 \times 512$ | $3 \times 3$ | Final Conv | $512 \times 512 \times 2$ | $1 \times 1$ |
| | Conv-Set 5 | $32 \times 32 \times 512$ | $3 \times 3$ | | | |

To explore the effectiveness of AMs, five of the best-known and most cited AMs in the RS domains are selected. These AMs are placed in the attention blocks at each training session. Results of the framework with AMs determine the AMs' effects quantitatively, but since AMs have complex structure and add many parameters to the network, they are not straightforward to understand and interpret. Therefore, the framework utilizes XAI methods to qualitatively examine the impact of the AMs. Two layer-based XAI methods, including Layer Gradient X activation and Layer DeepLIFT, were used as explainable gates before and after each attention block to analyze the features before and after passing the attention blocks.

In the following, a brief description of attention modules and the XAI methods is presented.

*3.1. Attention Methods*

Fundamentally, the human visual system does not process entire scenes at the same time. Instead, we focus on salient and selected parts of a scene to capture some form of visual composition [68]. This innate vision mechanism is the core inspiration of the AM, where focus is placed on the most significant information rather than putting equal attention on all available information. In DL, AM is used to focus on specific features, areas, or components of an image. This mechanism is generally divided into two groups/networks: channel and spatial attention [33]. The channel attention network strengthens channels in feature maps that transmit more valuable information, while suppressing those that contain less valuable information. The spatial attention network, on the other hand, highlights key regions in the feature space while ignoring background regions. Both attention networks can be utilized either separately or in unison within DL architectures to emphasize relevant feature layers and the respective region of interest.

3.1.1. Squeeze-and-Excitation Networks (SE)

One of the most popular AMs is the SE approach [27], which is presented in Figure 2A. SE is made up of three parts: (i) squeeze module, which reduces the spatial dimension to a scalar by global average pooling, (ii) excitation module, which learns the adaptive scaling weights for the channels by multi-layer perceptron (MLP), and (iii) scale module, where the "*excited*" tensor is passed through a sigmoid activation layer to scale the value to be between 0 and 1. Eventually, the output is directly multiplied element-wise with the input tensor.

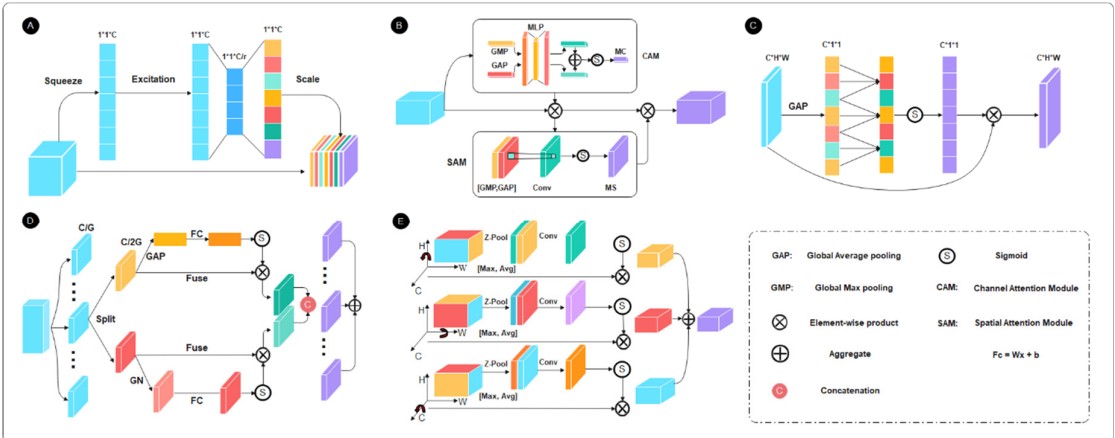

**Figure 2.** The architecture of utilized AMs, including (**A**) SE, (**B**) CBAM, (**C**) ECA, (**D**) Shuffle, (**E**) Triplet attention. In each training session, one of these AMs is placed into 4 attention blocks.

### 3.1.2. Convolutional Block Attention Module (CBAM)

The 3D attention map can be separated into two independent processes instead of being directly computed. Basically, separate attention generation has lower computational and parameter overhead, making it suitable for usage with pre-existing base CNN architectures. To build an attention map, CBAM [28], as presented in Figure 2B, is separated into two sequential components called the channel attention module (CAM) and the spatial attention module (SAM). CAM is a one-dimensional map, used for highlighting the important feature maps for learning, and is quite similar to the Squeeze Excitation layer. It produces a vector called channel weights, which is element-wise multiplied with each corresponding feature map in the input tensor. SAM, on the other hand, is a two-dimensional spatial attention map that concentrates on useful information in each feature map for learning. Combining these two has the potential of significantly increasing model performance.

### 3.1.3. Efficient Channel Attention (ECA)

To some extent, ECA is akin to that of the SE module. The ECA-net adopts a local neighborhood size, represented by the value k, for each tensor where channel attentions are computed within k, with respect to every other channel. ECA-net, as illustrated in Figure 2C, is composed of three parts: (i) global feature descriptor by using GAP, (ii) adaptive neighborhood interaction, and (iii) broadcasted scaling. The main difference between SE and ECA is the second part, in which the reduced tensor is subjected to 1-D striding convolution and the kernel size is determined by Equation (1).

$$k = \varphi(c) = \left|\frac{\log_2(C)}{\gamma} + \frac{b}{\gamma}\right|_{odd} \tag{1}$$

where $k$ is the local neighborhood kernel size, $b$ and $\gamma$ are predefined hyper-parameters set to be 1 and 2, respectively, and $C$ is the global channel space.

### 3.1.4. Shuffle Attention (SA)

SA takes its inspiration from the trinity of SE channel attention, spatial attention from CBAM, and channel grouping from spatial group enhance (SGE). SA, as shown in Figure 2D, is composed of four components: (i) feature grouping, which divides the input tensor into groups along the channel dimension, which is carried out before passing these groups to the attention module (which eventually is split into two branches along the channel dimension); (ii) channel attention, where a GAP is used to reduce the input dimensions to a vector and produces channel attention weights through a gating mechanism; (iii) spatial

attention to where the group norm (GN) is used over the input to obtain spatial-wise statistics (ultimately a compact feature representation is created through a gating mechanism, producing spatial attention weights); and (iv) aggregation, in which a channel shuffle strategy is adopted to allow cross-group information flow along the channel dimension.

### 3.1.5. Triplet Attention

The triplet attention module is concerned with the computation of attention weights by capturing cross-dimension interaction through a three-branch structure [29]. As illustrated in Figure 2E, it comprises three parallel branches. Given channel dimensions *C*, the first two branches capture cross-dimension interaction between C and the H (horizontal axis) or W (vertical axis) dimension by using rotation along H and W axes. The final branch determines/builds simple spatial attention, which is akin to CBAM [28]. At the very end, outputs from the branches are combined through averaging, forming the triplet attention output.

### *3.2. Explainable Artificial Intelligence (XAI)*

An explanation is a way to verify the output decision made by an AI agent or algorithm. Thus, the explainability of ML algorithms has become a pressing issue. DNNs have achieved remarkable success in real-world applications in various engineering fields [69], as well as in RS. However, the large number of parameters in DNNs make them complex to understand and undeniably harder to interpret. Regardless of evaluation parameters, which might indicate good learning performance, DL models could inherently succeed or fail to learn representations from the data that a human might consider important. Explaining the decisions made by DNNs requires knowledge of their internal operations. To discover this, XAI methods can be used to provide human-interpretable explanations of ML/DL black-box decisions. In this article, two layer-wise XAI methods (i.e., Layer Gradient X activation and Layer DeepLIFT) are used to explore the effect of AMs in RS domain.

Layer Gradient X activation performs element-wise multiplication of the layer's activation with the gradients of the target output, with respect to the given layer. It is a combination of the gradient and activation methods of layer attribution. Layer DeepLIFT on the other hand, is a recursive prediction explanation method for DL. It is a back-propagation-based approach that attributes a change to inputs based on the differences between the inputs and corresponding references (or baselines) for non-linear activations. As such, DeepLIFT seeks to explain the difference in the output from reference in terms of the difference in inputs from reference. It considers both positive and negative contribution scores of each layer.

## 4. Experiments

### *4.1. Dataset*

In general, DL-based methods have a large number of parameters to learn, which means they require massive amounts of high-quality data during the training and validation processes. This fact remains true even when an extra module such as an attention block is added. In this work, we are therefore using the xBD dataset [32], which is the largest and highest-quality satellite RS imagery dataset for building segmentation and damage assessment. The xBD dataset covers 19 different locations around the world and includes buildings of many shapes and types. Moreover, the samples are taken from densely populated urban areas, which adds a useful level of complexity to the dataset. In addition, this will also help us assess the model's performance in extracting multi-scale targets.

Figure 3 shows the coverage of the xBD dataset, which comprises 22,068 images. There is a total of 850,736 building polygons from 19 different locations, covering 45,000 km². The dataset contains image pairs of pre- and post-disaster images of size 1024 × 1024

pixels with three spectral bands (red, green and blue—RGB), with a resolution of 0.8 m/pixel. In our experiments, we used pre-disaster images for building segmentation. The dataset is organized into two tier sets consisting of tiers 1 and 3. The tier 1 set contains pre-disaster images subdivided into train, test, and holdout partitions (for algorithm training). The tier 3 set contains additional data without partitions. We use train and tier 3 partitions for train, test, and hold partitions for test and validation, respectively.

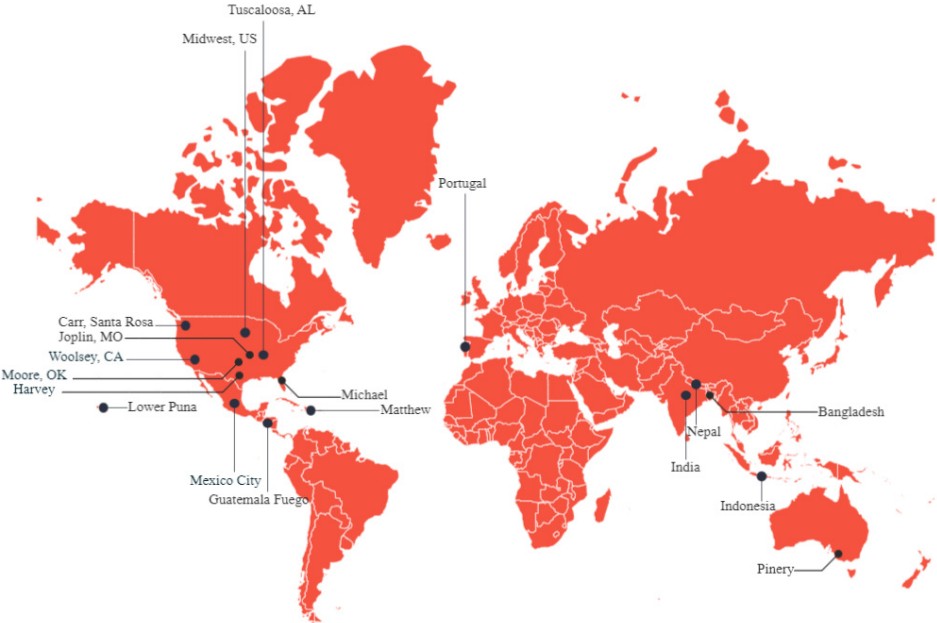

**Figure 3.** The locations from which data in the xBD dataset are collected [32].

### 4.2. Implementations Details

Experiments are conducted using a dual Intel Xenon E5-2698 v4 CPU and eight Tesla V100 GPUs by parallel-distributed data processing (DDP). For model input during training, the images are resized to $512 \times 512$ pixels for both training and test images. Binary-cross-entropy is used as the loss function, which is formulated as Equation (2), where $L$ quantifies the overall loss by comparing the target label $y^i$ with predicted label $\hat{y}^i$ and $N$ is the number of training pixels. The optimizer used is Adam (Adaptive Moment estimation) with a learning rate of $1.5 \times 10^{-4}$. The networks are trained for 50 epochs with a batch size of 32.

$$L = -\sum_{i}^{N} y^i \log \hat{y}^i + (1 - y^i) \log(1 - \hat{y}^i) \tag{2}$$

### 4.3. Results Analysis

Accuracy Assessment

For primary quantitative model evaluation, five metrics were selected:

(i) Precision—the ratio of correctly predicted buildings to the total number of samples predicted as buildings.
(ii) Recall—the proportion of correctly predicted buildings among the total buildings.
(iii) F1-score—computed as the harmonic mean between precision and recall.
(iv) IoU—measures the overlap rate between the detected building pixels and labeled building pixels (ground truth).

(v) Overall Accuracy (OA)—the ratio of the number of correctly labeled pixels to the total number of pixels in the whole image.

In this section, the quantitative summary of the vanilla network vs. the attention-based network is presented. For this, metrics that can be obtained from the confusion matrix are shown in Table 2. Note that the numbers in brackets denote the performance improvement over the vanilla model. We look mainly at F1-score and OA, as both these metrics provide an overall view of how the network performs.

**Table 2.** Building segmentation with different backbones with different AMs. The best record and improvement in each metric are marked in bold. The numbers in the brackets are the difference between the results of attention-based network and vanilla network.

| Backbone | Metric | Vanilla | SE | CBAM | ECA | SA | Triplet |
|---|---|---|---|---|---|---|---|
| SegNet | Precision | 87.03 | 89 (1.97) | **89.31** **(2.28)** | 88.04 (1.01) | 87.77 (0.74) | 88.02 (0.99) |
| | Recall | 75.32 | 74.98 (-0.34) | 75.59 (0.27) | **77.83** **(2.51)** | 75.46 (0.14) | 75.4 (0.08) |
| | OA | 97.87 | 97.96 (0.09) | 98.01 (0.14) | **98.06** **(0.19)** | 97.92 (0.05) | 97.93 (0.06) |
| | IoU | 67.72 | 68.63 (0.91) | 69.32 (1.6) | **70.39** **(2.67)** | 68.29 (0.57) | 68.38 (0.66) |
| | F1-score | 80.75 | 81.39 (0.64) | 81.88 (1.13) | **82.62** **(1.87)** | 81.15 (0.4) | 81.22 (0.47) |
| Unet | Precision | 87.07 | 88.76 (1.69) | 88.9 (1.83) | 87.53 (0.46) | **89.67** **(2.6)** | 88.42 (1.35) |
| | Recall | 81.06 | 85.05 (3.99) | 84.5 (3.44) | **85.69** **(4.63)** | 82.55 (1.49) | 85.46 (4.4) |
| | OA | 98.16 | **98.47** **(0.31)** | 98.45 (0.29) | 98.42 (0.26) | 98.4 (0.24) | **98.47** **(0.31)** |
| | IoU | 72.36 | 76.79 (4.43) | 76.44 (4.08) | 76.37 (4.01) | 75.38 (3.02) | **76.86** **(4.5)** |
| | F1-score | 83.96 | 86.87 (2.91) | 86.64 (2.68) | 86.6 (2.64) | 85.96 (2) | **86.91** **(2.95)** |

From Table 2, it is clear that the SegNet-based framework with AM showed improvements based on most of the measured metrics, as compared to vanilla. In particular, ECA and CBAM achieved better results compared to other AMs and vanilla. Best improvements in F1-score are by ECA and CBAM, which improved by 1.87% and 1.13%, respectively. In addition, CBAM improved precision by 2.28%. Based on the other metrics, ECA improved the recall by 2.51%, IoU by 2.67%, and OA by 0.19%, respectively.

In the U-net frameworks, the AMs show mixed results, but the improvements seem more noticeable. The triplet achieved the best results with an F1-score improvement by 2.9%, OA by 0.31%, and IoU by 4.5%. Additionally, the SE attention result was similar to triplet and is better than the other attentions. Other attentions results are acceptable and generally show F1-score improvements of ~2%.

All in all, it seems that quantitatively, especially for F1-score, overall performance increased with AMs regardless of the CNN backbone. For AMs such as CBAM and ECA on SegNet, and SE and Triplet on Unet, F1-score improves between ~2% and ~3%, respectively. Moreover, for recall and IoU, the improvements for SegNet and Unet backbones are ~2.5% and ~5%, respectively.

*4.4. Attention Analysis by XAI*

Analysis of the AMs' effectiveness is performed through attribution visualizations of the XAI methods. Specifically, the feature tensors before and after passing the attention blocks in each layer can be examined. Figures 4 and 5 show attribution visualization results for the AMs based on two-layer attribution models. The visualizations attempt to intuitively demonstrate that pixels (features) with higher attribution are considered "*more important*" for predicting a class. Basically, more important pixels have a darker shade of green, whereas less important pixels move towards the darker shade of red. To aid understanding, we include an attribution value scale at the base of Figure 4.

| Proposed framework based on SegNet | | | | | | |
|---|---|---|---|---|---|---|
| Attention | XAI Layer | Situation | 1st | 2nd | 3rd | 4th |
| SE | Layer Gradient X Activation | Before Attention |  |  |  |  |
| | | After Attention |  |  |  |  |
| | DeepLIFT | Before Attention |  |  |  |  |
| | | After Attention |  |  |  |  |
| CBAM | Layer Gradient X Activation | Before Attention |  |  |  |  |

| | | | | | | |
|---|---|---|---|---|---|---|
| | | After Attention | | | | |
| | DeepLIFT | Before Attention | | | | |
| | | After Attention | | | | |
| ECA | Layer Gradient X Activation | Before Attention | | | | |
| | | After Attention | | | | |
| | DeepLIFT | Before Attention | | | | |
| | | After Attention | | | | |
| Shuffle | Layer Gradient X Activation | Before Attention | | | | |
| | | After Attention | | | | |
| | DeepLIFT | Before Attention | | | | |

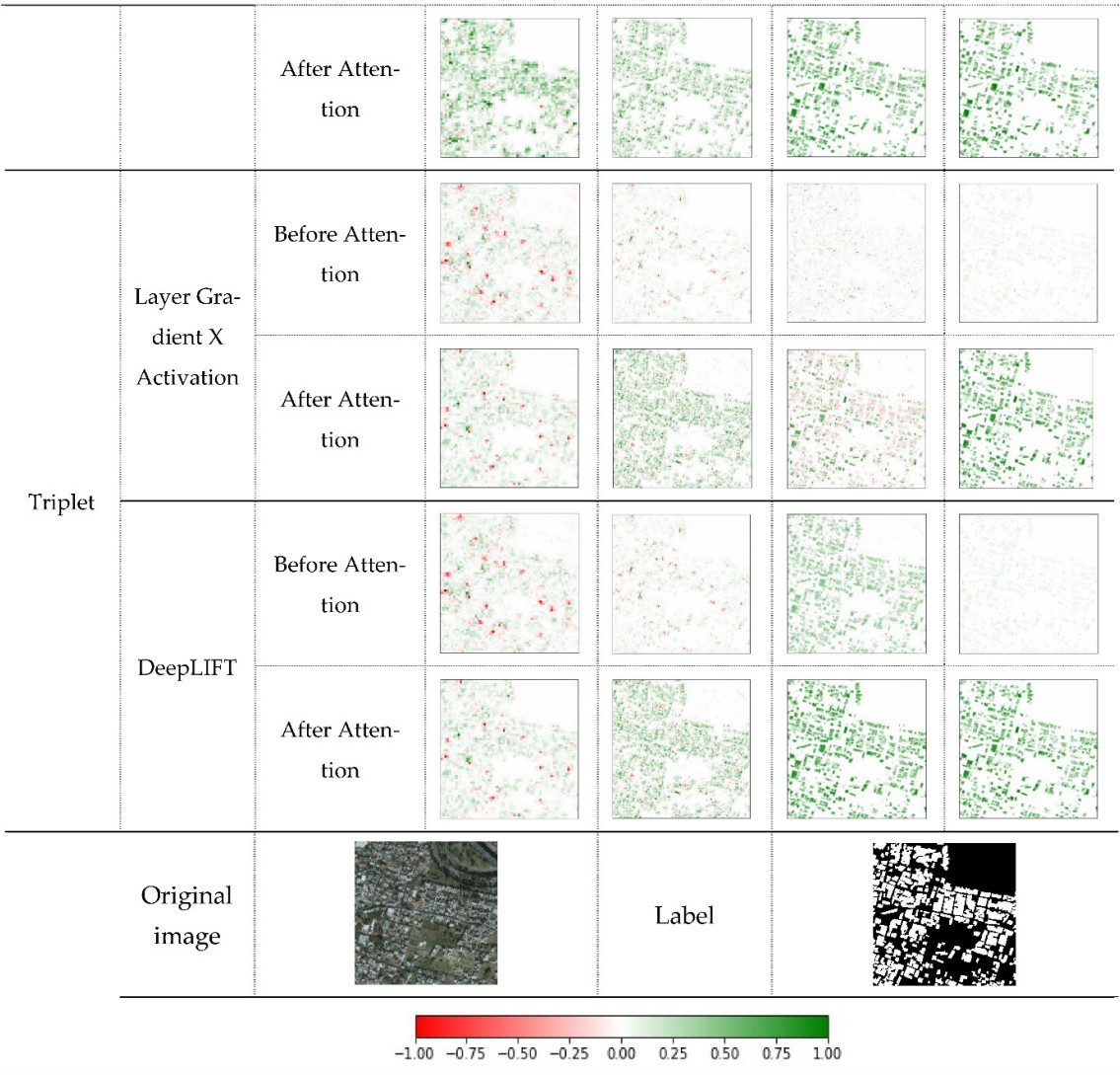

**Figure 4.** Visualization results of Gradient X Activation and LayerDeepLIFT before and after AMs on SegNet. The numbers above the table (i.e., 1st, 2nd…) indicate the layer number of the attention in the framework.

| Proposed framework based on Unet | | | | | | |
|---|---|---|---|---|---|---|
| Attention | XAI Layer | Situation | 1st | 2nd | 3rd | 4th |
| SE | Layer Gradient X Activation | Before Attention |  |  |  |  |
| | | After Attention |  |  |  |  |
| | DeepLIFT | Before Attention |  |  |  |  |
| | | After Attention |  |  |  |  |
| CBAM | Layer Gradient X Activation | Before Attention |  |  |  |  |

| | | After Attention |  |  |  |  |
|---|---|---|---|---|---|---|
| | DeepLIFT | Before Attention |  |  |  |  |
| | | After Attention |  |  |  |  |
| ECA | Layer Gradient X Activation | Before Attention |  |  |  |  |
| | | After Attention |  |  |  |  |
| | DeepLIFT | Before Attention |  |  |  |  |
| | | After Attention |  |  |  |  |
| Shuffle | Layer Gradient X Activation | Before Attention |  |  |  |  |
| | | After Attention |  |  |  |  |
| | DeepLIFT | Before Attention |  |  |  |  |

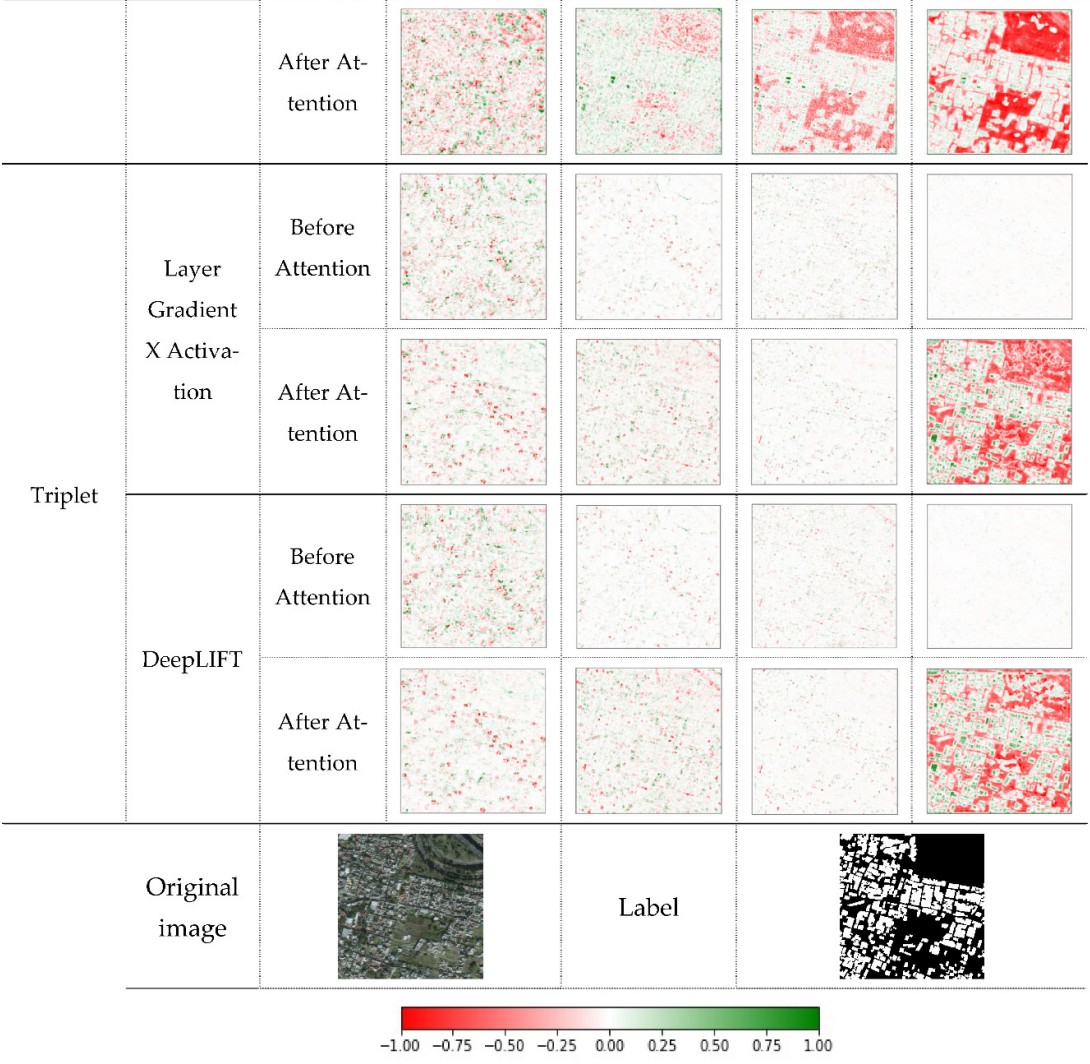

**Figure 5.** Visualization results of Gradient X Activation and LayerDeepLIFT before and after AMs on Unet. The numbers above the table (i.e., 1st, 2nd…) indicate the layer number of the attention in the framework.

The SegNet-based framework's attribution visualization is shown in Figure 4. Both layer attributions reveal that SE heavily focuses on the building objects, especially in the last layers where the network is trying to decide. The attribution values also show that SE attempts to distinguish between background and building object in the earlier levels of the attention. Then the network deepens, and we can see a reduction in the influence of the irrelevant objects such as trees and the street as well as other background objects. Also shown is absolute attribution (high response value—dark green) with deeper network depth, which indicates the effectiveness of the SE mechanism.

Based on the visualization for DeepLIFT and Gradient, with the deepening of the network, CBAM is able to highlight the building object as the network target and reduces the influences of the background noise, especially in the two last layers of the attention. A closer look at the visualization reveals that the network is not able to clearly focus on the building before attention in most layers. However, CBAM in the 3rd and 4th layer manages to highlight buildings and achieve high attribution values as the main target. This is also reflected in the model evaluation metrics improvements.

Visualization of the attribution value shows some difference between the input and output tensors to the attention. It also depicts the network being able to comprehensively mark the building as an important area while ignoring irrelevant objects. All the images further prove the effectiveness of the ECA and good performance of the network in detecting buildings. Furthermore, higher attribution values (darker) after the 2nd attention layer prior to the attention indicate good performance. The results of the attention layer attribution of SA are similar that of ECA. There are some differences between input and output to the attention. Overall, though, the images show that attention does make the network focus more on the buildings.

These results provide a rough indication that the attention leads the network to concentrate more on the buildings, as depicted in the pixels with higher attribution. This entails "higher importance" in predicting that class. The earlier levels of the attention seem to demonstrate the process of separating buildings from background and other irrelevant objects. This is indicated by the lower attribution values (red). We can also see that as the network deepens, the attention mechanism forces the network to focus on the buildings as the main target. All in all, it could be concluded that with the deepening of the network, the effect of attention mechanism becomes quite obvious.

Figure 5 shows the attribution visualization of the proposed framework based on Unet. Upon close observation of the SE attention visualization results, the earlier stages of the attention attempt to distinguish between the buildings and background and irrelevant objects (shown in after attention visualization with higher attribution values in first two layers). With the deepening of the network, the effectiveness of the attention seems reduced, and the mechanism is not able to properly distinguish the target.

The early levels of CBAM were unable to distinguish between the background and the target buildings. There was, however, some difference between input and output tensors of the attention; the last layer of the attention seems to fully highlight the buildings as the target, while reducing the effect of background on the network decisions. Based on the attribution values, the effect of ECA on the input tensor in the early stages is bigger than the last layer. As the network goes deeper, the effect of the attention becomes reduced, and attention begins to focus on irrelevant regions of the image instead of the buildings.

Shuffle attention seems to heavily focus on the building as the main target and reduce the effect of the other irrelevant region on the network. The AMs successfully mark urban areas that are actual building areas as important regions. After the attention, the influence of irrelevant regions, such as trees and streets, reduced.

The attribution values of the output tensor of the attention are a good reason for the effectiveness of triplet. The visualizations after attention show pixels (features) with higher attribution are considered "more important" in predicting that class. The pixels with the highest attribution values are the pixels belonging to buildings. Moreover, at the earlier levels, the attention tries to distinguish between buildings and background, while in the last layers, the attention highlights buildings as the main target of the network.

For further analysis, the effect of the AMs on the last layer is explored by unboxing the effects of the attentions on the last layer of the networks. This is performed using LayerDeepLIFT. Several areas from the dataset were fed to the vanilla and attention-based network, and the results are presented in Figures 6 and 7.

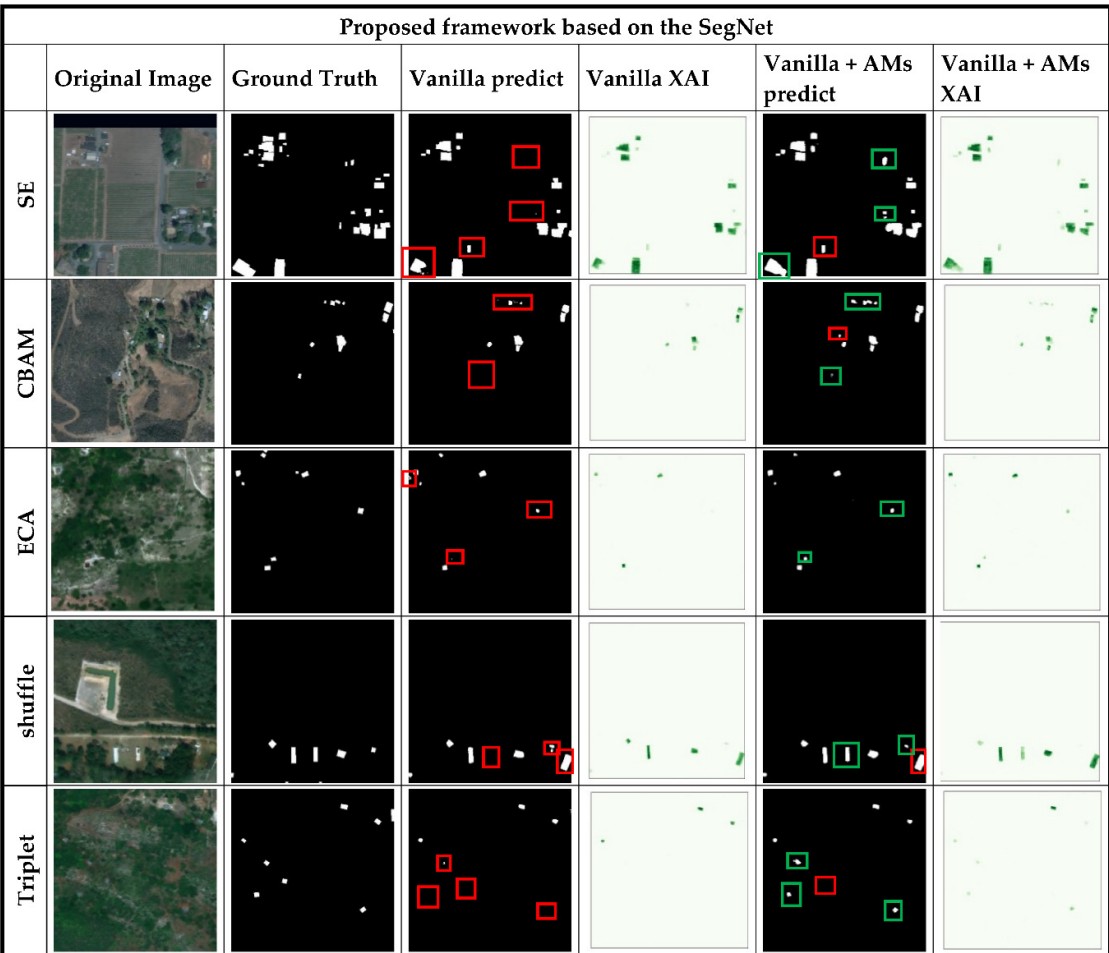

**Figure 6.** Visualization of predicted labels for vanilla SegNet and attention-based SegNet and LayerDeepLIFT (red box: missed building and wrongly labeled building, green box: true building). In XAI outputs, related objects should be green and bolder than other objects.



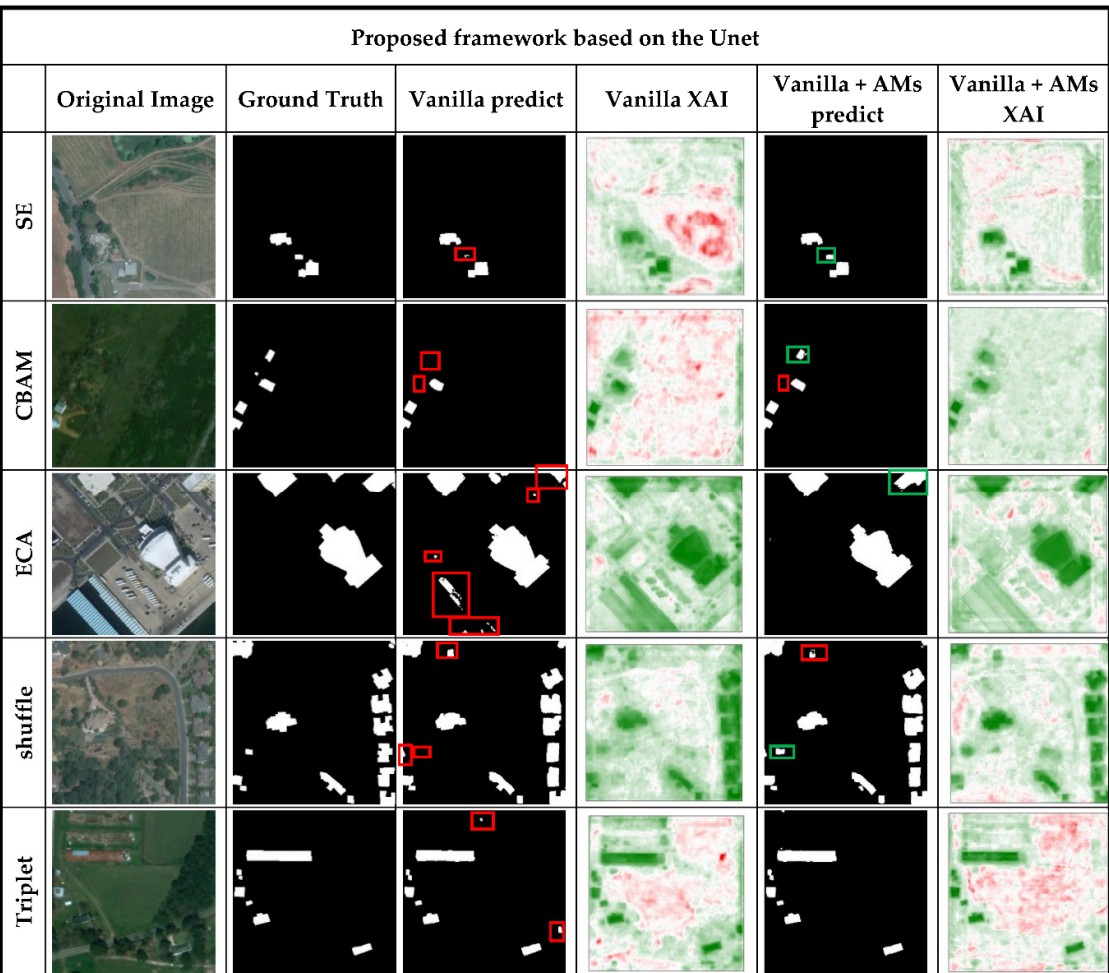

**Figure 7.** Visualization of predicted label for vanilla Unet and attention-based Unet and Lay-erDeepLIFT (red box: missed building and wrongly labeled building, green box: true building). In XAI outputs, related objects should be green and bolder than other objects.

A close look at Figure 6 shows that all predicted outputs of the attention-based networks have better building segmentation results. In certain cases, it is clearly shown in the attribution visualization, that although the buildings are small, the attention-based networks of LayerDeepLIFT of the last layer are still superior in building segmentation compared to the vanilla networks. Comparing LayerDeepLIFT's visualization of the vanilla and attention-based networks roughly shows that the attention-based networks heavily focus on the small building regions. In addition, some of the detected buildings in the vanilla network are incomplete. Notably, the shape of the detected buildings in the attention-based network has more similarity with the ground truth, especially with regards to the buildings' boundaries. Moreover, detected buildings in the attention-based network have higher attribution values than the vanilla, which indicates that the AMs motivate the network to concentrate on the buildings.

Figure 7 presents the predicted output of Unet's and LayerDeepLIFT's visualization of the last layers of the networks. Close observation shows that the vanilla network produces more false positive pixels. However, almost all the attention-based networks perform better and properly delineate non-building objects. Moreover, the higher attribution values of the visualization of LayerDeepLIFT indicate better performance of the attention-based network, especially in the output of the ECA, in which the vanilla network labeled

large non-building objects as building, and ECA improve the performance and labeled non-building objects as background.

### 4.5. Computational Analysis

As AMs consist of trainable layers and filters, additional parameters need to be estimated. This inevitably adds more computational cost. To analyze the efficiency of AMs, we compare parameter size, computational cost, floating point operations per second (FLOPs), and inference time in comparison to the vanilla network (based on SegNet) with all attentions. As shown in Table 3, most of the attention blocks include less than 0.01 million additional parameters, and therefore have roughly the same computational cost. SE and CBAM add a slight increase in computational cost in respect to other attentions. We also calculated the average inference time for each image ($512 \times 512 \times 3$ pixels) in the testing phase. The inference time for all attentions is roughly the same, and the maximum additional run time is related to CBAM attention (16.72 ms), which is negligible.

**Table 3.** Complexity comparison in terms of number of blocks, network parameters, inference time, and FLOPs. Comparing these terms indicates how much complexity AMs add to vanilla network.

| Model | Vanilla | SE | CBAM | Triplet | SA | ECA |
|---|---|---|---|---|---|---|
| Number of blocks | - | 4 | 4 | 4 | 4 | 4 |
| Parameters (k) | - | 87 | 44.98 | 1.2 | 0.528 | 0.012 |
| Total (Million) | 20.74 | 20.83 | 20.78 | 20.74 | 20.74 | 20.74 |
| Inference time (ms/img) | 23.60 | 23.81 | 40.32 | 25.94 | 24.40 | 23.79 |
| FLOPs (GMac) | 110.44 | 110.44 | 110.44 | 110.46 | 110.45 | 110.44 |

### 4.6. Transferability of the Proposed Framework

To verify the robustness and transferability of the proposed framework, the WHU building segmentation dataset (East Asia sub-dataset) [70] is used. This dataset consists of images from 6 neighboring satellites, totaling 17,388 high-resolution 512 px × 512 px images, covering 860 km$^2$ with 0.45 m ground resolution. From these images, 13,662 tiles are used for training and 3726 tiles for testing. We initially present the quantitative summary of the vanilla network (without attention) vs. the network with attention. Next, three of the best AMs for each backbone are selected to be further evaluated, for which the resulting quantitative metrics are presented in Table 4. It can be clearly seen from Table 4 that all the selected AMs improve the performance of both backbones compared to vanilla. However, the improvements are different from that of the xBD dataset results.

**Table 4.** Building segmentation with different backbones and AMs on WHU building segmentation dataset. The best record and improvement in each metric are marked in bold. The numbers in the brackets are the difference between the results of attention-based network and vanilla network.

| Backbone | Metric | IoU | Recall | Accuracy | Precision | F1-Score |
|---|---|---|---|---|---|---|
| SegNet | Vanilla | 60.07 | 67.53 | 99.43 | 84.47 | 75.06 |
| | SE | 65.26 | 74.65 | 99.50 | 83.85 | 78.98 |
| | | (5.19) | (7.12) | (0.07) | (-0.62) | (3.92) |
| | CBAM | **67.28** | **75.18** | **99.54** | 86.49 | **80.44** |
| | | **(7.21)** | **(7.65)** | **(0.11)** | (2.02) | **(5.38)** |
| | ECA | 61.97 | 67.23 | 99.48 | **88.79** | 76.52 |
| | | (1.9) | (-0.3) | (0.05) | (4.32) | (1.46) |
| Unet | Vanilla | 63.15 | 72.89 | 99.46 | 82.53 | 77.41 |
| | SE | 67.15 | 75.32 | 99.53 | **86.09** | 80.35 |
| | | (4) | (2.43) | (0.07) | **(3.56)** | (2.94) |
| | CBAM | 68.07 | **79.48** | 99.53 | 82.59 | 81.00 |
| | | (4.92) | **(6.59)** | (0.07) | (0.06) | (3.59) |

| | | | | |
|---|---|---|---|---|
| Triplet | **68.73** **(5.58)** | 79.31 (6.42) | **99.54** **(0.08)** | 83.75 (1.22) | **81.47** **(4.06)** |

As with the xBD dataset, similar analysis was performed on the selected AMs for the WHU dataset results on the backbones (i.e., attribution visualizations of the XAI methods). Figure 8 shows the attribution visualization results for the selected AMs based on two-layer attribution models. Both XAI layer attributions in SegNet reveal that all the AMS heavily focus on building objects, especially in the last layers where the network performs decision. A closer look at the visualization reveals minor differences between input tensors of AMs and output tensors of AMs. In some cases, however, such as CBAM, they are able to highlight the building object at the end of the network and improve network performances. Moreover, the attribution visualizations of Unet reveal that the AMs focus on delineating between background and building and distributing more attention to buildings as the main target of the network.

| Proposed framework based on SegNet | | | | | | |
|---|---|---|---|---|---|---|
| Attention | XAI Layer | Situation | 1st | 2nd | 3rd | 4th |
| SE | Layer Gradient X Activation | Before Attention | | | | |
| | | After Attention | | | | |

| | | | | | | |
|---|---|---|---|---|---|---|
| | DeepLIFT | Before Attention | | | | |
| | | After Attention | | | | |
| CBAM | Layer Gradient X Activation | Before Attention | | | | |
| | | After Attention | | | | |
| | DeepLIFT | Before Attention | | | | |
| | | After Attention | | | | |
| ECA | Layer Gradient X Activation | Before Attention | | | | |
| | | After Attention | | | | |
| | DeepLIFT | Before Attention | | | | |
| | | After Attention | | | | |

**Proposed framework based on Unet**

| | | | | | | |
|---|---|---|---|---|---|---|
| SE | Layer Gradient X Activation | Before Attention | | | | |
| | | After Attention | | | | |
| | DeepLIFT | Before Attention | | | | |
| | | After Attention | | | | |
| CBAM | Layer Gradient X Activation | Before Attention | | | | |
| | | After Attention | | | | |
| | DeepLIFT | Before Attention | | | | |
| | | After Attention | | | | |
| Triplet | Layer Gradient X Activation | Before Attention | | | | |
| | | After Attention | | | | |

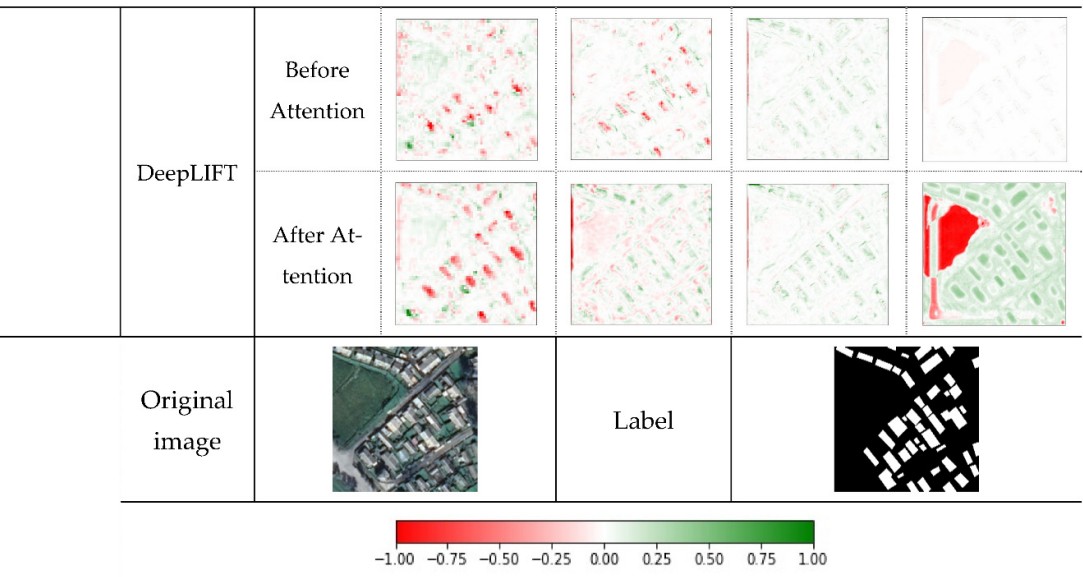

**Figure 8.** Visualization results of Gradient X Activation and LayerDeepLIFT before and after AMs on SegNet and Unet for WHU dataset. The numbers above the table (i.e., 1st, 2nd…) indicate the layer number of the attention in the framework.

## 5. Conclusions

In this work, the effectiveness of AMs in RS big data are analyzed based on the five metrics of F1-score, OA, recall, precision, and IoU. The objective was to determine whether AMs play any significant role when applied to RS tasks. Our results quantitatively show that AMs do improve overall performance, albeit on selected network backbones (i.e., SegNet and Unet) and the specific type of AMs used. Investigation into model interpretability using the two XAI methods, CBAM and SE, revealed interesting results. With CBAM, the two last blocks of attention show focus on building as the main target, while separating background from building objects. In SE, the attention mechanism seems to perform gradual building segmentation as the network goes deeper. We have also observed that the XAI methods reveal the internal layer decisions of the networks, which greatly facilitate model interpretability by humans.

Overall, we measured the performance of the attention-based networks using DeepLIFT. The results support our assumption that attention-based networks improve detection of small buildings without adding much computational complexity. False positives were also reduced compared to results when using a vanilla network. In general, the novelties of this paper are twofold; first, the effectiveness of the AMs in RS related task are evaluated and interpretations of attention blocks in different layers of the framework are provided, and second, the black box of the DL models is unboxed using XAI layer contribution methods.

For future work, we will observe the impact of the XAI results from this work. Specifically, we will explore their uses in the design and arrangement of different blocks, in tuning the number of input and output channels, determination of kernel size, and other tuning network parameters.

**Authors' Contributions:** E.H.Z., L.M., and B.K. acquired the data; E.H.Z., L.M., and B.K. conceptualized and performed the analysis; E.H.Z. and L.M. wrote the manuscript and discussed and analyzed the data; B.K. supervised; B.K. and N.U. provided the funding acquisition; B.K., N.U., and A.A.H. provided technical sights, as well as edited, restructured, and professionally optimized the manuscript. All authors have read and agreed to the published version of the manuscript.

**Funding:** The APC is supported by the RIKEN Centre for AIP, Tokyo, Japan.

**Data Availability Statement:** The datasets that support the findings will be available at https://xview2.org/ (09/12/2022) and http://gpcv.whu.edu.cn/data/building_dataset.html (09/12/2022).

**Acknowledgments:** The authors would like to thank RIKEN Centre for AIP, Japan, for providing all facilities during the research.

**Conflicts of Interest:** The authors declare no conflict of interest.

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
