# Peer review of "Unboxing the Black Box of Attention Mechanisms in Remote Sensing Big Data Using XAI"

_remotesensing, doi:10.3390/rs14246254_

Round 1
Reviewer 1 Report (Previous Reviewer 3)
The revision has addressed my previous concerns. I would like to raise my rating to acceptance.
Author Response
Dear Reviewer,
Thank you for your positive comment.
Best Regards,
Dr. Kalantar
Reviewer 2 Report (New Reviewer)
Overall, the concept as well as the manuscript is very well organized. Apart from some minor adjustments in text and improvement in quality of figures, the paper is suitable for publishing.
Author Response
Dear Reviewer,
Thank you for reviewing the paper. We have improved the quality of figures.
Best Regards,
Dr. Kalantar
This manuscript is a resubmission of an earlier submission. The following is a list of the peer review reports and author responses from that submission.
Round 1
Reviewer 1 Report
The authors propose a framework exploring the effectiveness of AMs (Attention Mechanisms) in tasks of Remote Sensing (RS) imaging procedures. They used to evaluate the proposed methodology, the xBD dataset, specifically images with buildings and the task was to discriminate this type of objects. The detection methodology is based on Convolutional Neural Network (CNN) architectures and they compare methods designing experiments with and without AMs, for a first study. Different quantitative metrics (five) were included in this study. Complementing this research, the authors exploring the Explainable Artificial Intelligence (XAI) methods. Layer gradient X activation and layer DeepLIFT are used to investigate the internal AMs and the overall effects of the AMs on the network. In this case, qualitative evaluation is based on color-coded value attribution to assess how the AMs facilitate the CNNs in performing buildings classification. The results seem to show that AMs improve the quantitative metrics.
I find the paper to be interesting, the procedures are well planned and well described. The contribution seems specified in the “Related works” part and I consider it a valuable contribution to the Remote Sensing journal.
Nonetheless, I have some concerns:
1) Line 15. Is necessary to include in this line the description of AMs. Nevertheless it is in the line 17.
2) Line 20. It is was better add the word N to vanilla. By the way, in some cases “vanilla” and in other cases “vanila”….
3) Different lines. The author should unify the description of acronyms, the first with capital letters?....
4) Line 31, 32. Can you explain this: ……with the attribution visualization results of XAI methods agreeing with the quantitative metrics.
5) Line 502. Can you explain (in short) what is the resize image procedure from 1024 to 512?
6) Line 524-525. In this section, the quantitative summary of the vanilla network (without attention) vs. with attention is presented….. (without attention) is reiterated in different lines of the text.
7) The conclusions should provide highlighting the novelties brought by your proposed framework and why your propose is better than the others.
Author Response
Dear Reviewer
We would like to thank you for careful and thorough reading of this manuscript
and for the thoughtful comments and constructive suggestions, which help to improve the quality of this manuscript. Our responses are attached.
Best Regards,
Dr. Bahareh Kalantar

Reviewer 2 Report
Paper shows the impact of using attention mechanism in remote sensing segmentation task. The results show that attention can improve the efficiency of the models. The improvement is explained by using two chosen XAI methods. The formatting and language in many places should be improved (e.g. tables, table 2 is not table 1, table 3 not table 2, maybe native speaker should review it). Maybe second remote sensing dataset should be used to proof the effect more strongly.
'By analyzing of 728 the XAI results, the arrangement of the different block, number of input 729 and output channel, kernel size and other parameters can be tuned' - maybe discussion about that should be earlier in the text, and more should be added about it,
somewhere in a text it is mentioned that U-Net is deeper than Segnet, in Table 1 it is shown that after adaption to your testing framework they have the same number of layers.
more discussion should be added about attention mechanisms and different result between them, why they have different XAI results, what can be the reason of that
Author Response
Dear Reviewer
We would like to thank you for careful and thorough reading of this manuscript
and for the thoughtful comments and constructive suggestions, which help to improve the quality of this manuscript. Our responses are attached.
Best Regards,
Dr. Kalantar

Reviewer 3 Report
Summary:
This paper studies the effectiveness of Attention Mechanisms (AM) in remote sensing big data. The authors proposed an encoder-decoder CNN framework for building segmentation; based on that, they evaluated the effectiveness of AMs from both predictive performance and attribution explanations perspectives.
Pros:
-
The authors conducted extensive experiments to evaluate the effectiveness of AMs.
-
The paper structure is well organized, and the Related Studies are comprehensive.
Cons:
-
Experiments on a single dataset. The experiments only use the xBD dataset, although it’s “the largest and highest quality satellite RS imagery dataset”, there is a risk that the conclusions found on this dataset may not consistent with other large-scale datasets.
-
Suggestion: it would be better if you could add one more dataset to demonstrate your conclusion are generalized.
-
Different images from several areas make it hard to compare between AMs. During analyzing the effect of AMs on the last layer of the networks (Figures 6 & 7), different images from several areas are provided for different AMs. Although it still demonstrates the difference between w/ and w/o a certain AM, it is hard to compare the effect of different AMs.
-
Suggestion: it would be better if you could use the same image (as you did in Figures 4 & 5) to make comparisons between different AMs as well.
-
The last layer explanations of Unet often highlight the border of the images. For both vanilla and vanilla+AM, the explanations of Unet often highlight the border of the images (Figure 7). This looks problematic and it may indicate problems during model training. For example, a possible reason, did you pad the image with 0s or other values?
-
Suggestion: it would be better if you could explain this phenomenon, or provide convincing proof that your training protocol is correct.
Addition comments on writing:
-
The Related Study. It is comprehensive, but it’s more like listing the existing works, hard to follow and finding the gap between the proposed work and existing ones (although you have a conclusion paragraph at the end of this section).
-
Suggestion: I personally think it would be better if you could not only list the existing works, but also identify the gap between the proposed work and the listed ones at the end of each paragraph.
-
Figure/Table captions can be more informative. Typically, we do not want to miss any important information in the figure/table captions. The most efficient way to understand your paper is to read your figures, so it is important to make them self-explained.
-
Suggestion: Good captions include two main components: 1) what is the figure is about; 2) the main takeaway from this table/figure.
-
(Minor typo). There are two Table 1, the second one should be Table 2.
Author Response
Dear reviewer,
We would like to thank you for careful and thorough reading of this manuscript
and for the thoughtful comments and constructive suggestions, which help to improve the quality of this manuscript. Our responses are attached.
Best Regards,
Dr. Kalantar
